# Protein primary structure correlates with calcium oxalate stone matrix preference

Yu Tian[1]*, Matthew Tirrell[1]*, Carley Davis[2]*, Jeffrey A. Wesson[3]**

**1** Pritzker School of Molecular Engineering, University of Chicago, Chicago, Illinois, United States of America, **2** Department of Urology, Department of Veterans Affairs Medical Center and the Medical College of Wisconsin, Milwaukee, Wisconsin, United States of America, **3** Department of Medicine/Nephrology, Department of Veterans Affairs Medical Center and Medical College of Wisconsin, Milwaukee, Wisconsin, United States of America

* These authors contributed equally to this work.

* jwesson@mcw.edu

## Abstract

Despite the apparent importance of matrix proteins in calcium oxalate kidney stone formation, the complexity of the protein mixture continues to elude explanation. Based on a series of experiments, we have proposed a model where protein aggregates formed from a mixture containing both strongly charged polyanions and strongly charged polycations could initiate calcium oxalate crystal formation and crystal aggregation to create a stone. These protein aggregates also preferentially adsorb many weakly charged proteins from the urine to create a complex protein mixture that mimics the protein distributions observed in patient samples. To verify essential details of this model and identify an explanation for phase selectivity observed in weakly charged proteins, we have examined primary structures of major proteins preferring either the matrix phase or the urine phase for their contents of aspartate, glutamate, lysine and arginine; amino acids that would represent fixed charges at normal urine pH of 6–7. We verified enrichment in stone matrix of proteins with a large number of charged residues exhibiting extreme isoelectric points, both low (pI<5) and high (pI>9). We found that the many proteins with intermediate isoelectric points exhibiting preference for stone matrix contained a smaller number of charge residues, though still more total charges than the intermediate isoelectric point proteins preferring the urine phase. While other sources of charge have yet to be considered, protein preference for stone matrix appears to correlate with high total charge content.

## Introduction

Calcium oxalate (CaOx) kidney stone disease is an increasingly prevalent disease [1–3], but its pathogenesis remains poorly defined [4, 5]. While disease activity correlates with the calculated supersaturation of urine with respect to calcium oxalate crystal formation based on the small ion chemistry, the urine supersaturation is a poor predictor of stone forming disease state in randomly selected patients [5]. Consequently, research over many decades has focused on studies of proteins, which comprise a small, but ubiquitous and necessary component in

**Data Availability Statement:** Original proteomics data were previously published in Urolithiasis in two separate articles (see references 8 and 21 as described in our Methods section). While these articles were initially published as restricted access

for the first year, they were made fully available (free access) at the end of one year in accordance with requirements from the NIH and US government employee rules. These data were supplemented with amino acid content data from the widely used and freely accessible "Uniprot" database (www.uniprot.org). Accession numbers for this database are available in the Supporting Information files. All new data used in these analyses are also included in the Supporting Information files.

**Funding:** Direct support for this work was supplied by the Department of Veterans Affairs through Merit Review funding from the Clinical Sciences Research and Development Division (grant number CX001491; JAW PI) and through the use of resources and facilities at the Clement J Zablocki Department of Veterans Affairs Medical Center, Milwaukee, Wisconsin. Also, indirect support was provided by a National Institutes of Health/ National Institute for Diabetes, Digestive, and Kidney Diseases (grant number DK82550; JAW PI).

**Competing interests:** The authors have declared that no competing interests exist.

kidney stones [6, 7]. Initially, these studies focused on various individual proteins without finding a consistent link between any protein and disease state. Many studies in the past 10 years have demonstrated the presence of hundreds of unique proteins in CaOx stone matrix using proteomics methods [8–13], but the sheer number of proteins involved has confounded efforts to define their role in stone formation.

We have advanced the hypothesis that the formation of protein aggregates plays a critical role in the initiation of CaOx stone formation [8, 14]. The potential link between protein aggregate formation and risk for stone formation was established in two *in vitro* studies; one showing that the protein aggregates formed by mixing strong polyanions with strong polycations would induce CaOx crystal aggregation [14], and another showing that self-aggregates of uromodulin (a prevalent anionic protein in urine and stone matrix) would induce CaOx crystal aggregate formation [15]. Recently published data from quantitative proteomics experiments have shown that the stone matrix contains many urine proteins, but that the relative abundances of these proteins differ substantially from their abundance in urine for most proteins [8]. Furthermore, the enrichment of strongly anionic proteins and strongly cationic proteins in stone matrix supports the postulate that aggregation of these strongly ionic proteins was the triggering event, with most other proteins exhibiting partitioning behavior between the matrix and urine phases [8, 14, 16].

Proteins containing anionic side chains, particularly carboxylic acid side chains, were expected to be important based on their frequent observation in stone matrix, as well as clear evidence of their ability to influence CaOx crystal formation in the laboratory [14, 17, 18]. The generally inhibitory properties of strongly anionic proteins on CaOx crystallization *in vitro* seems contradictory to the frequently observed presence of these same proteins in stone matrix [8–12], but repeated observation of their presence in matrix indicates that such proteins were likely critical to the buildup of stones. Strongly cationic proteins would likely aggregate with strongly anionic proteins driven by electrostatic attraction between the oppositely charged polymers (proteins); a process that has been studied extensively in synthetic polymers [19, 20]. Other, more weakly charged proteins are drawn to such aggregates in a selective manner that mimics their selectivity toward stone matrix, as observed in a model system where polyarginine was added to the normal mixture of proteins found in urine [16]. The physical chemical basis of this phase selectivity has not been described to date.

In this study, we have examined and compared the amino acid composition in both the most abundant proteins exhibiting preference for stone matrix and the most abundant proteins exhibiting preference for the urine phase. The goal of this comparison is to identify the primary structural features that determine protein preference for one phase over the other, as well as to explore more fully the details of our hypothesis stated above. We have specifically enumerated amino acid residues in each protein that would represent fixed charges at normal urine pH's of 6–7 (aspartate (D), glutamate (E), lysine (K), and arginine (R)) to better understand the contribution of this primary structural feature to the observed protein distributions in clinical stone samples.

## Materials and methods

### Protein characteristics

We identified stone matrix preferring proteins (MPPs) and urine preferring proteins (UPPs) from our previously reported proteomics analysis results. Based on the comparison of the mean protein abundances from 8 CaOx monohydrate (COM) stone matrix samples to 25 stone former urine (SFU) samples [8], 7 proteins, including UROM (uromodulin), EGF (epidermal growth factor), APOD (apolipoprotein D), ALBU (albumin), IGKC (Ig kappa chain C

region), IGHG1 (Ig gamma-2 chain C) and IPSP (plasma serine protease inhibitor), from the list of top 60 COM matrix proteins (Table 2 in [reference 8]) were found at higher relative abundance in SFU samples. Therefore, these 7 proteins were added to a list of 31 abundant proteins from SFU (relative abundance > 0.5%) that were not found in the MPP list [21], to create a list of UPPs. The remaining COM matrix proteins in the list from the top 60 COM matrix proteins are thus identified as MPPs. Due to updates of protein databases, the detailed protein primary information of the originally identified ubiquitin was not unambiguously defined. Therefore, after excluding ubiquitin, a total of 52 MPPs are enumerated with their amino acid compositions in S1 Table, and 38 UPPs are enumerated with their amino acid compositions in S2 Table.

## Ethics statement

This Ethics statement was added in response to concerns raised in review. All protein identity and relative abundance data used in this analysis were obtained from previously published manuscripts [references 8 and 21].

Protein identity and abundance data for SFU samples were extracted from Table 2 and Supplementary Data Table S6 [reference 21], and these data were sorted as above. Patients were recruited for this study, and written informed consent was obtained under the oversight of the Clement J Zablocki IRB#1 (study number 9305-01P). All data were de-identified (only gender and age included) prior to publication.

COM matrix preferring protein identities and abundances were extracted from Table 2 [reference 8], and these data were sorted as above. Four CaOx kidney stones in this study were obtained from de-identified (except for age and gender), pathological waste specimens previously characterized at the Mandel International Stone and Molecular Analysis Center (MIS. MAC, Milwaukee, WI, USA) or the National VA Crystal Identification Center (Milwaukee, WI, USA) and were used without obtaining IRB approval. An additional four stones were obtained from newly recruited patients with written informed consent from stone removal surgery under the oversight of the Medical College of Wisconsin/Froedtert Hospital Institutional Review Board monitoring (protocol number PRO21952), and these samples were also de-identified, only race, age, and gender included.

## Protein amino acid content

The amino acid content of D, E, K and R residues for each intact protein were determined for both groups of proteins based on primary structural data in the Uniprot database (www. uniprot.org). The number of D, E, K, and R residues for each protein are listed in S1 and S2 Tables (for MPPs and UPPs respectively), along with the calculated values defined below. Using these data and assigning a charge value of 1 for each of these amino acids, the total charge residues in each protein was readily calculated as the sum of D+E+K+R, though the total charge percentage (total charge/total residue number x 100%) provides a more fair comparison of individual protein properties (normalizing for molecular weight). The net charge for each protein was also calculated as (K+R)-(D+E), so that the appropriate sign (negative vs positive) was represented for each protein, and these data are also included in S1 and S2 Tables, along with the net charge percentage, which was calculated in a similar manner (net charge/total residue number x 100%).

Two-tailed nonparametric Spearman's correlation coefficient was computed for the correlation analysis of net charge percentage and net charge number of MPPs. Two-tailed Mann-Whitney test was used to compare total charge percentage of MPP and UPP groups. Statistical analysis was performed using GraphPad Prism.

## Results

The main comparisons of these protein groups were made on the basis of charge percentage data for two reasons. First, the percentage of charge residues provides a more realistic measure of average protein or polymer properties for these charged proteins than does the total, because it normalizes for large differences in molecular weights between proteins. Second, while most proteins in either category fall in the molecular weight range of 25,000 Da and 100,000 Da, the few proteins with much higher molecular weights with correspondingly larger numbers of charged residues would cause to distortion of graphical representations by compressing most other proteins into a narrow region of the graph. Clear differences in the distribution of proteins between MPP and UPP groups are evident in cluster plots of total charge percentage vs net charge percentage, as shown in Fig 1. Clearly the UPPs exhibit much tighter distributions of both total charge percentage and net charge percentage, than do the MPPs. As expected, most proteins in either group were negatively charged, even without considering post translational modifications (glycosylation and phosphorylation), which would tend to make these proteins more negatively charged. In addition, the proteins exhibiting extreme net charge percentage values, either positive or negative, clearly manifest larger total charge percentage values, though there were a few proteins with nearly zero net charge percentage that had larger total charge percentage in the MPP group. We tested the statistical relevance of this association using a two-tailed Spearman's correlation analysis, as shown in Fig 2 in a plot of net charge percentage vs net charge number for the MPPs. A strong correlation coefficient $r = 0.923$ ($p < 0.0001$) between extreme total charge and extreme net charge was confirmed.

A more direct comparison of net charge percentage distributions between MPP and UPP groups is illustrated in cluster plots in Fig 3. This plot emphasizes the presence of both strongly net positively charged proteins and strongly net negatively charged proteins in stone matrix. Following the tenets of the proposed model, these MPPs with extreme charge are obvious and can easily be removed in a further analysis to examine differences between the weakly charged MPPs and the similarly charged UPPs. The MPP list in S1 Table was truncated by removing proteins with either net charge percent > 5% or net charge percent < - 5%, eliminating 15 proteins (8 anionic ones and 7 cationic ones) from the MPP list to generate a truncated MPP table

### Matrix preferring proteins (MPPs)   Urine preferring proteins (UPPs)

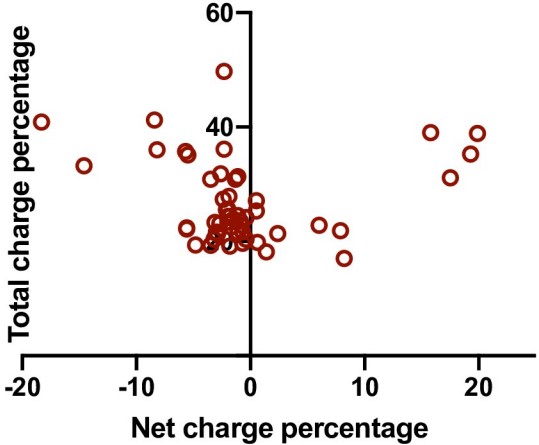
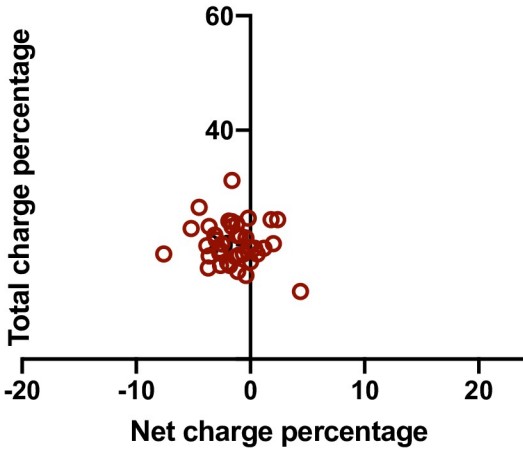

**Fig 1. Cluster plots of total charge percentage vs. net charge percentage.** Matrix preferring proteins (MPP) are shown in the left hand panel, and urine preferring proteins (UPP) are shown in the right hand panel. Many more proteins with both high total charge percentage and extreme net charge percentage were observed in the MPP set than in the UPP set.

## Spearman's correlation

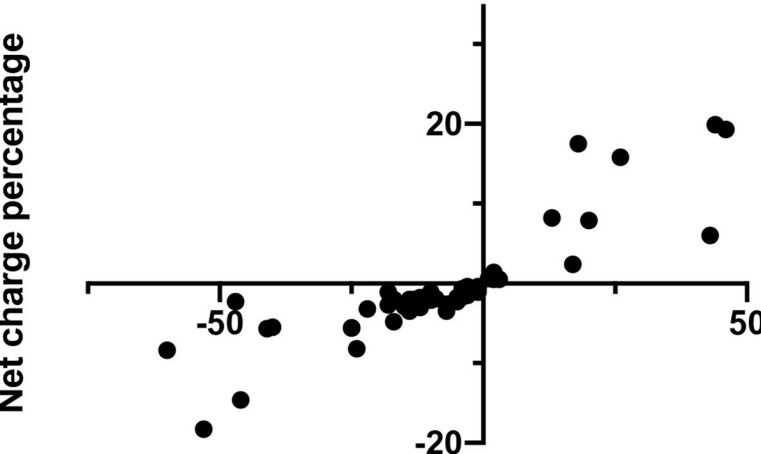

**Fig 2. Spearman's correlation analysis plot.** The net charge percentage was plotted against the net charge number for MPPs in this figure demonstrating a strong correlation (r = 0.923; p<0.0001).

shown in S3 Table. No adjustments were made to the urine preferring protein list, though 3 proteins remaining in the UPP list fell outside of these cutoffs by small amounts.

Using the truncated MPP set, the comparisons were then made between both the net charge percentage and the total charge percentage parameters, as shown in Fig 4. The first and third data columns show the net charge percentage distributions, while the corresponding total charge percentage distributions are plotted in columns two and four. The net charge percentage distributions are indistinguishable, but the weakly charged MPPs generally contained a larger percentage of total charges. A two-tailed Mann-Whitney test reveals a statistically significant ($p < 0.0001$) larger total charge percentage in the weakly charged (net) MPPs (median 23%, n = 37 or mean 25±6%) than in UPPs (median 20%, n = 38 or mean 20±4%). Note that in the truncated MPP set, one protein, HMGB1 (high mobility group protein B1), shows a much larger total charge percentage of 49.7% than most other proteins, though it is still exhibited a low net charge percent of -2.3%. Removing HMGB1 from this analysis did not affect the outcome, as the total charge percent difference between stone matrix and urine-preferred proteins remained highly statistically significant ($p < 0.0001$).

## Discussion

The analysis of amino acid contents of matrix preferring proteins has confirmed the expected characteristic that the proteins at either extreme of isoelectric point distribution (pI either <5 or >9) contained a relatively large total number of charge groups compared to other matrix proteins or proteins that prefer the urine phase in patient samples. Polyelectrolyte theories certainly would predict that these oppositely charged proteins would be most strongly attracted to one another by electrostatic interactions based on the respective net charge percentages, which conceptually supports the idea that these proteins with extreme net charge percentage trigger aggregation [19, 20]. Once these aggregates are formed, they clearly also create a microenvironment that is enriched in charged residues of either sign. The presence of such aggregates is

# Net charge percentage

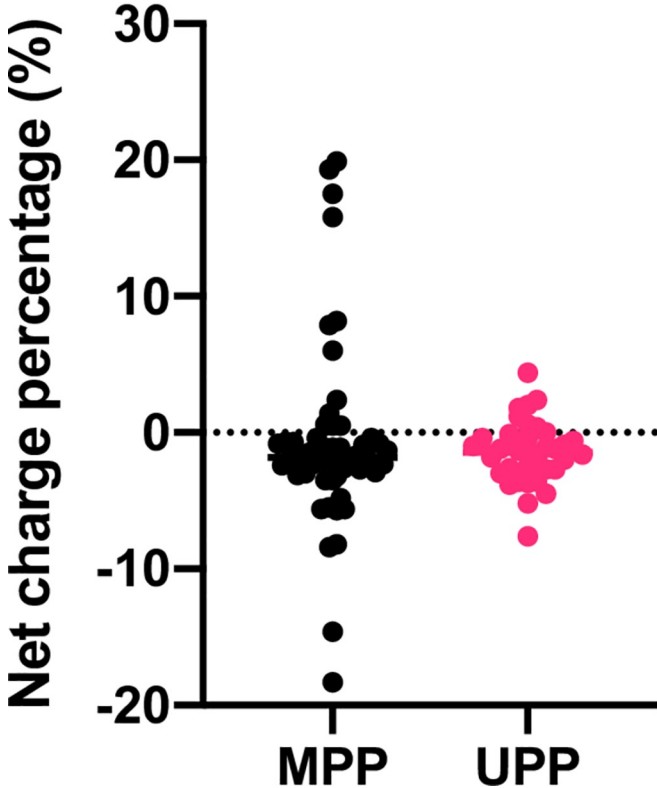

**Fig 3. Cluster plots of the net charge percentage.** Both MPPs (black dots) and UPPs (pink dots) are represented here, with the MPP distribution demonstrating many more proteins with extreme values of net charge percentage (both positively and negatively charged).

correlated with in vitro effects on crystallization phenomena, particularly inducing aggregation of COM crystals [14, 22], consistent with the formation of stones. Unfortunately, laboratory experiments to date have not explored the alteration of protein-crystal interactions that might derive from inclusion of many more weakly charged proteins in these mixtures that would be representative of actual stone matrix, nor has consideration been given to alterations of the protein-protein interactions that might support the protein aggregate stability.

The largest number of MPPs share similar characteristics to UPPs, in that their net charge percentage is near zero, and they are more weakly charged overall than proteins with extreme isoelectric points. An important new finding from this analysis is elucidated in the comparison of the total charge percentage between the truncated set of MPPs made by removing proteins with extreme isoelectric points and urine preferring proteins. In this comparison, the total charge content in MPPs was still greater than that observed in UPPs with a high level of certainty. This finding implies that ionic interactions rather other hydrophobic interactions are the most important feature directing phase selectivity of weakly charged proteins for the matrix phase. Furthermore, the fact that protein sorting between aggregate and solution phase in the previously published model system of poly-arginine (pR) induced protein aggregates from urine protein mixtures [16] mimicked the distribution observed in patient COM stone matrix samples [8] suggests that this protein selectivity is the result of protein-protein

## Net and total charge percentage analysis

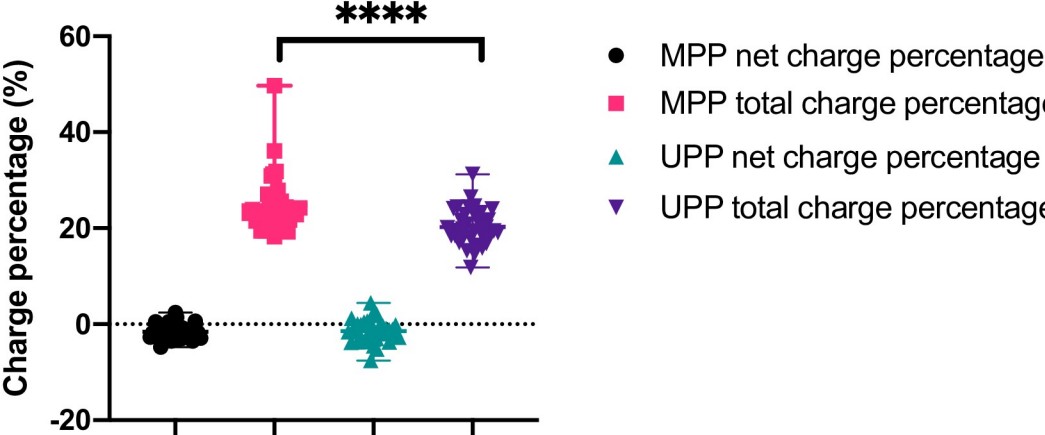

**Fig 4. Cluster plots of net charge percentage and total charge percentage comparing the truncated MPP set to UPP.** Net charge percentage clusters for both the truncated set of MPP (black circles) and UPP (green triangles) were tightly clustered around 0, as expected, with no difference observed between the mean values. The total charge percentage cluster for MPP data (pink squares) demonstrated a broader distribution extending particularly to higher total charge values with a larger mean value than the UPP data (purple inverted triangles); 25±6% vs 20±4% respectively (p<0.0001).

interactions rather than protein-crystal interactions. The larger total charge contents of the near zero net charge matrix preferring proteins is also intuitively satisfying, because of their greater similarity to total charge values exhibited by proteins with more extreme isoelectric points that are thought to trigger aggregate formation.

This analysis is limited by several factors. First, as with all mass spectral studies of protein distributions, the presence of intact proteins is imputed from the observations of selected fragments of those proteins, though limited gel electrophoresis data support the presence of intact proteins [14, 23]. Second, very few proteomics studies on stone disease and urine have been designed to report both identity and relative abundance of all proteins observed. Most studies have focused on the more typical search for "marker" proteins. While the lists of proteins identified in matrix and urine are largely overlapping, the analysis in this study can only be applied to a few of these studies [8, 9]. Third, the level of variation of protein distributions over time in patients has not been well characterized in either stone formers or healthy adults. More specifically, no data exist for urine proteomics during a time of active stone growth in stone formers, though it seems unlikely that variations large enough to explain the differences between stone matrix and urine protein distributions could have been missed in the examination of random samples from the dozens of stone formers reported to date. Fourth, charge contributions from other sources, such as glycosylation and phosphorylation, have not been considered. In general, these protein modifications would add negative charges to proteins and increase total charge percentages. Many proteins found in matrix and urine are glycoproteins and have phosphorylation sites, but glycosylation levels are difficult to measure across the entire distribution of proteins by mass spectrometry in complex mixtures, such as these. Phosphorylation levels can more easily be obtained, but these measurements would require a complete re-analysis of mass spectral data sets, which was beyond the scope of this study. Experiments exploring the impact of more complicated protein mixtures on crystallization phenomena would appear to be more important at this stage to more fully characterize stone formation.

## Conclusions

In conclusion, examination of the amino acid contents of D, E, K, and R in proteins showing selective preference for COM stone matrix revealed 3 groups of proteins that support the mechanism of stone formation we have proposed. Proteins in the first 2 groups are characterized by both large net charge percentage (one is negatively charged and the other is positively charged) and large total charge percentage, and these proteins would likely aggregate at very low concentrations, potentially triggering stone formation. The third group contains a number of individual proteins with near zero net charge percentage and somewhat lesser total charge percentage. When compared to proteins showing preference for the urine phase in patient samples, the matrix preferring proteins exhibit a similar distribution of net charge percentages, but significantly higher total charge percentage than the urine preferring proteins, suggesting the preference for stone matrix is linked with high total charge residue contents. The behavior of these more complicated mixtures of proteins in crystallization assays deserves further attention.

## Supporting information

**S1 Table. Stone matrix phase preferring protein list.** The list of the top 60 most highly abundant and highly prevalent proteins found in CaOx stone matrix (from Table 2 in [reference 8]) was reduced by removing 7 proteins, including UROM (uromodulin), EGF (epidermal growth factor), APOD (apolipoprotein D), ALBU (albumin), IGKC (Ig kappa chain C region), IGHG1 (Ig gamma-2 chain C) and IPSP (plasma serine protease inhibitor) that were found at higher relative abundance in urine than in stone matrix. Additionally, ubiquitin was also removed from this list due to an ambiguous definition of this protein in updated protein databases, leaving 52 proteins that were found to be enriched in CaOx stone matrix in [Reference 8]. The abbreviations, accession numbers, and protein names are given for each protein, as well as the number of aspartate (D), glutamate (E), arginine (R), and lysine (K) residues in each protein, listed as both absolute totals and percentages for each amino acid. Total charge and net charge were calculated for each protein as described in the Methods section, and each was listed as both absolute totals and percentages for these 52 proteins. A column containing updated accession numbers for 3 proteins which had changed from the original publication has been added.
(XLSX)

**S2 Table. Urine phase preferring protein list.** The 7 proteins from the list of the top 60 most highly abundant and highly prevalent proteins in CaOx stone matrix (from Table 2 in [Reference 8]) that demonstrated higher abundance in urine than CaOx stone matrix, including UROM (uromodulin), EGF (epidermal growth factor), APOD (apolipoprotein D), ALBU (albumin), IGKC (Ig kappa chain C region), IGHG1 (Ig gamma-2 chain C) and IPSP (plasma serine protease inhibitor), were added to the 31 proteins found in stone former urine at high relative abundance ($>0.5\%$) but not in the list of the top 60 proteins from CaOx stone matrix in SFU samples (see Supplementary Data Table S6 in [Reference 21] and Table 2 from [Reference 8]) to create the list of 38 urine preferring proteins in this table. The abbreviations, accession numbers, and protein names are given for each protein, as well as the number of aspartate (D), glutamate (E), arginine (R), and lysine (K) residues in each protein, listed as both absolute totals and percentages for each amino acid. Total charge and net charge were calculated for each protein as described in the Methods section, and each was listed as both absolute totals and percentages for these 38 proteins. A column containing the updated accession number for 1 protein which had changed from the original publication has been added.
(XLSX)

**S3 Table. Truncated matrix phase preferring protein list.** The MPP list in S1 Table was truncated by removing the 15 highly charged proteins (8 anionic ones with charge $< -5\%$ and 7 cationic ones with charge $> +5\%$), and the truncated list is presented together with the UPP list from S2 Table. Only the protein abbreviations, total charge percentages, and net charge percentages are included in this table.
(XLSX)

## Author Contributions

**Conceptualization:** Matthew Tirrell, Carley Davis, Jeffrey A. Wesson.

**Data curation:** Yu Tian, Jeffrey A. Wesson.

**Formal analysis:** Yu Tian, Jeffrey A. Wesson.

**Funding acquisition:** Jeffrey A. Wesson.

**Investigation:** Yu Tian, Matthew Tirrell, Carley Davis, Jeffrey A. Wesson.

**Methodology:** Yu Tian, Matthew Tirrell, Jeffrey A. Wesson.

**Project administration:** Jeffrey A. Wesson.

**Software:** Yu Tian.

**Supervision:** Matthew Tirrell, Jeffrey A. Wesson.

**Writing – original draft:** Yu Tian, Matthew Tirrell, Jeffrey A. Wesson.

**Writing – review & editing:** Yu Tian, Matthew Tirrell, Carley Davis, Jeffrey A. Wesson.

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
