## [Decision Letter · Decision Letter 0]

18 Jun 2021

PONE-D-21-15604

Protein primary structure correlates with calcium oxalate stone matrix preference

PLOS ONE

Dear Dr. Wesson,

Thank you for submitting your manuscript to PLOS ONE. After careful consideration, we feel that it has merit but does not fully meet PLOS ONE’s publication criteria as it currently stands. Therefore, we invite you to submit a revised version of the manuscript that addresses the points raised during the review process.

We look forward to receiving your revised manuscript.

Kind regards,

Sanjay Kumar Singh Patel, Ph.D.

Academic Editor

PLOS ONE

Journal Requirements:

2. Thank you for including your ethics statement.

a. Please amend your current ethics statement to include the full name of the ethics committee/institutional review board(s) that approved your study.

3. In your ethics statement in the manuscript and in the online submission form, please provide additional information about the patient samples described as "Regarding CaOx stone matrix proteomics data reported in reference 8, four CaOx kidney stones in this study were obtained from de-identified, pathological waste specimens previously characterized at the Mandel International Stone and Molecular Analysis Center (MIS.MAC, Milwaukee, WI, USA) or the National VA Crystal Identification Center (Milwaukee, WI, USA) and were used without obtaining IRB approval." Specifically, please ensure that you have discussed whether all samples were fully anonymized before you accessed them.

"We gratefully acknowledge the primary financial support provided through the Department of

 Veterans Affairs Merit Review funding (CX-001491; JAW) along with resources and the use of

facilities at the Clement J. Zablocki Department of Veterans Affairs Medical Center, Milwaukee,

WI, and in part by the National Institutes of Health/National Institute for Diabetes, Digestive, and

Kidney Diseases for earlier direct funding of this work (DK 82550) (JAW)."

"JAW is principal investigator on Merit Review CX001491, funded by Department of Veterans Affairs, Clinical Sciences Research and Development Division, which supported this work. "

7. Your ethics statement should only appear in the Methods section of your manuscript. If your ethics statement is written in any section besides the Methods, please move it to the Methods section and delete it from any other section. Please ensure that your ethics statement is included in your manuscript, as the ethics statement entered into the online submission form will not be published alongside your manuscript.

Reviewers' comments:

Reviewer's Responses to Questions

**Comments to the Author**

1. Is the manuscript technically sound, and do the data support the conclusions?

Reviewer #1: Yes

Reviewer #2: Yes

2. Has the statistical analysis been performed appropriately and rigorously? 

Reviewer #1: I Don't Know

Reviewer #2: Yes

3. Have the authors made all data underlying the findings in their manuscript fully available?

Reviewer #1: Yes

Reviewer #2: Yes

4. Is the manuscript presented in an intelligible fashion and written in standard English?

Reviewer #1: Yes

Reviewer #2: Yes

5. Review Comments to the Author

Reviewer #1: Authors are proposing an interesting hypothesis about the initial steps of the stone formation. They are suggesting that protein-protein interaction precedes protein ions/crystal interaction. Th question is why protein protein interaction is favored over an interaction between say calcium and various calcium binding proteins/lipids. Lipids are considered to play a role in crystal nucleation and may also be in aggregation. Is there a change in urinary profile of stone formers that supports this phenomenon or is it a common occurrence in the urine. If protein contents of the stone formers urine is different from those of normal individuals, what is the source behind such a change.

May I suggest that you provide the full names of the proteins, when first used in the text.

Reviewer #2: The manuscript entitled “Protein primary structure correlates with calcium oxalate stone

matrix preference” by Tian et al tried to examined and compared the amino acid composition in both the most abundant proteins exhibiting preference for stone matrix and the most abundant proteins exhibiting preference for the urine phase. They concluded that protein preference for stone matrix appears to correlate with high total charge content However, this reviewer feels that the manuscript needs some changes.

Comments:

1. Did authors find these proteins in non-stone former urine samples?

2. Are these proteins specific to CaOx crystal stones only or they may be included in other stones matrix such as uric acid stones/ cystine stones?

3. What is the origin of these proteins, means diet origin/human origin?

4. What will be the proposed mechanism to control the stone formation and clinical relevance of this study?

5. What is the sequence of stone formation? Do proteins aggregate first followed by disposition of CaOx or vice versa?

6. Is stone formation at other organs also linked with this proposed mechanism? If yes, then include that data too.

6. PLOS authors have the option to publish the peer review history of their article (what does this mean?). If published, this will include your full peer review and any attached files.

Reviewer #1: No

Reviewer #2: **Yes: **Vinay Kumar

---

## [Author Response · Author response to Decision Letter 0]

1 Sep 2021

Specific comments to reviewer’s questions:

Reviewer 1:

“The question is why protein protein interaction is favored….” I have published several studies over many years focused specifically on characterizing the influence of proteins on calcium oxalate stone formation, and these experiments have supported the hypothesis that proteins likely assert primary influence over aggregation and growth of crystals (summarized in a review article, reference 14, in this manuscript). These studies have shown a 10,000 fold enhancement of inhibitory activities against nucleation, growth and aggregation of calcium oxalate crystal for selected relevant chemicals when comparing polymeric (30 or more residues bound in a chain structure) to monomeric forms of the same chemical, which is a direct justification for focus on the protein components. Proteins are also known to provide the dominant mass fraction to stone matrix, implying greater significance to their role in stone formation. This particular study was focused on identifying the underlying molecular interactions that likely lead to the collection of such a complex mixture of proteins in stone matrix; a mixture that is roughly duplicated by simply adding a strong polycation (polyarginine) to a solution of normal urinary proteins and studying the aggregates formed (reference 16 in this manuscript).

“Is there a change in urinary profile of stone formers…?” We have published a comparison the proteomic profiles in urine samples from healthy adults and stone formers (reference 21) finding relatively trivial differences between them, and certainly no evidence for the enrichment in stone former’s urine of either highly charged polyanions or highly charged polycations that we observed at increased abundance in stone matrix (reference 8). No studies have been published to date describing changes in the proteomic profile of urine proteins with time or environmental exposure that might help explain the stone matrix proteomic profile, though my current VA grant is funding research to look at urine proteomic profiles over time in both stone forming and healthy adults. Without an observed difference, we can only speculate on sources for a change in the urine proteome distribution.

We have provided the full names of proteins when they are first used in the text.

Reviewer 2:

1. The urine proteomic profiles of healthy adults and stone formers are almost completely overlapping/identical, but both urine profiles are different than what we observed in stone matrix (see references 8 and 21).

2. The majority of proteins observed in stone matrix in stones of all types are found in urine, but the relative amounts of individual proteins differ greatly between urine and matrix for calcium oxalate stones, as well as other stone types. Comparable data regarding the relative abundances of individual matrix proteins in other stone types have not been published yet. We do possess such data, and substantial differences can be seen between the stone matrix proteomic profiles observed in other stone types. At present, we cannot explain such differences, though we feel that they are important. We are still preparing this manuscript for publication.

3. Certainly, the proteins in human stones are of human origin, and cat stones contain cat proteins. Many urine proteins are secreted in the tubules, while others are filtered from the blood. Whether any of these proteins are derived from diet sources vs originating from normal physiologic processes within the body is currently unknown (unreported), and there are too many proteins involved in either urine or matrix to reasonably answer this question.

4. It is difficult to define a treatment before the mechanism is understood. Certainly, the presence of a large number of proteins in stone matrix that are normally found in intracellular or nuclear compartments argues for a cell injury process (likely tubular cells) as an originating event. We introduced this speculation in reference 8, but no further studies have been reported by us or others to address this possible mechanism, though hopefully my current VA funded investigation will provide some insight upon completion.

5. This question was addressed in the Discussion in reference 8. Current data cannot distinguish between a surge of highly charged polyanions and highly charged polycations creating protein aggregates which collect crystals vs a slow accumulation of these proteins from urine on existing crystals. The conversion of the very low abundance of the highly charged polyanions and polycations in urine to the observed distribution in stone matrix could be generated by the prolonged growth times of stones (months to years) through selective adsorption to crystal surfaces. A new experiment must be designed to test this question.

6. While I suspect that it is possible that pathological crystallization processes in other parts of the body may share protein characteristics with kidney stones, I am not aware of the appropriate protein data being available for crystal deposits in other organs to facilitate such a comparison.

I would like to publish the peer review process, since many questions have been posed and answered in the response to reviewers that may not fit in this manuscript specifically, but readers may find the discussions illuminating.

---

## [Editor Report · Decision Letter 1]

3 Sep 2021

Protein primary structure correlates with calcium oxalate stone matrix preference

PONE-D-21-15604R1

Dear Dr. Wesson,

We’re pleased to inform you that your manuscript has been judged scientifically suitable for publication and will be formally accepted for publication once it meets all outstanding technical requirements.

Kind regards,

Sanjay Kumar Singh Patel, Ph.D.

Academic Editor

PLOS ONE

---

## [Editor Report · Acceptance letter]

15 Sep 2021

PONE-D-21-15604R1 

Protein primary structure correlates with calcium oxalate stone matrix preference 

Dear Dr. Wesson:

I'm pleased to inform you that your manuscript has been deemed suitable for publication in PLOS ONE. Congratulations! Your manuscript is now with our production department. 

Kind regards, 

on behalf of

Dr. Sanjay Kumar Singh Patel 

Academic Editor

PLOS ONE